# Synchrotron "virtual archaeozoology" reveals how Ancient Egyptians prepared a decaying crocodile cadaver for mummification

Camille Berruyer [1,2] *, Stéphanie M. Porcier[3,4], Paul Tafforeau[1]

**1** European Synchrotron Radiation Facility, Grenoble, France, **2** ArchéOrient–UMR 5133, Maison de l'Orient et de la Méditerranée–Jean Pouilloux, Lyon, France, **3** Laboratoire CNRS "Archéologie des Sociétés Méditerranéennes"–UMR 5140, LabEx ARCHIMEDE, Université Paul-Valéry Montpellier 3, Montpellier, France, **4** Laboratoire CNRS HiSoMa–UMR 5189, Maison de l'Orient et de la Méditerranée–Jean Pouilloux, Lyon, France

* camille.berruyer@esrf.fr

## Abstract

Although Ancient Egyptians mummified millions of animals over the course of one millennium, many details of these mummification protocols remain unknown. Multi-scale propagation phase-contrast X-ray synchrotron microtomography was used to visualise an ancient Egyptian crocodile mummy housed at the Musée des Confluences (Lyon, France). This state-of-the-art non-destructive imaging technique revealed the complete interior anatomy of the mummy in three dimensions. Here, we present detailed insight into the complex post-mortem treatment of a decaying crocodile cadaver in preparation for mummification. Except for the head and the extremities of the limbs, everything beneath the skin of the crocodile (i.e. organs, muscles, and even most of the skeleton) was removed to cease further putrefaction. This unexpected finding demonstrates that earlier knowledge obtained from textual and other archaeological sources does not sufficiently reflect the diversity of mummification protocols implemented by Ancient Egyptians.

## Introduction

From the end of the Late Period (ca. 722–332 BC) into the Roman era (around the third century AD), Egyptians mummified millions of animals as part of their animal cult [1–7].

Ancient votive animal mummification was conducted using a wide range of protocols. A wealth of research into the topic and direct analyses of animal mummies, demonstrated that a large diversity of animal remains was preserved in these mummies. Some mummies contain a single animal (e.g. [8–10]). Of the 152 animal mummies analysed by L. McKnight to date, 86 (57%) contained a single complete individual [9, 11].

Votive animal mummies were not exclusively created from complete and well-preserved carcasses. Some bundles were made from several complete young individuals (especially in crocodiles mummies) [12]. Cases have been described where bundles contained detached body parts or parts of decaying animals [9,11–13]. Most of these mummies contain parts of a

**Data Availability Statement:** All the tomographic data used for the present study are available through the open access tomographic database of the ESRF (http://paleo.esrf.eu).

**Funding:** This research was carried out within the framework of the ESRF (ESRF-CFR-423), and the MAHES Project (Momies Animales et Humaines EgyptienneS) supported by the Agence Nationale de la Recherche through the «Investissement d'Avenir» program ANR-11-LABX-0032-01 LabEx ARCHIMEDE.

**Competing interests:** The authors have declared that no competing interests exist.

single specimen, although some of them preserve parts of several individuals of the same species, or elements of several individuals from different species [9, 11–13].

Nevertheless, to the extent of our knowledge, no detailed insights into the mummification techniques used for preserving largely putrefied cadavers have been reported. Crocodile mummy MHNL 90001850 presented here represents such a preparation protocol and reveals a thus-far unknown procedure for manufacturing votive mummies.

## Material

The studied crocodile mummy (MHNL 90001850; **Fig 1**) is housed in the Musée des Confluences, Lyon, France, no permit were required for the described study, which complied with all relevant regulations. The mummy is preserved in two parts. The anterior (head) part measures 21.1 cm and the posterior (body) part, lacking the tip of the tail, measures 40.4 cm. It is housed in the animal mummy collection of the Musée des Confluences in Lyon, France. This collection is the largest of its kind outside Egypt. The collection was gathered between 1897 and 1909, mainly by L. Lortet with assistance of G. Maspero [14–16]. Exploratory analyses on the collection (e.g. the unwrapping of mummies, isolating individual animals from mummies composed of multiple specimens, and cleaning of skeletons) were performed in the beginning of the XX[th] century by L. Lortet and C. Gaillard [16]. However, these analyses were either not properly recorded, or the documentation has been lost. Spatial and temporal provenance are unknown for MHNL 90001850, although L. Lortet and C. Gaillard mention that this specimen may have originated from either Esna or Kom Ombo [17]. Although radiocarbon dating may be used to indicate the age of the mummy, this destructive technique was not available to us within the scope of the present study.

Although M. Nicolotti and L. Robert (1994) already described MHNL 90001850, we made several observations conflicting their 1994 report. We therefore propose the following updated description. The mummy is completely unwrapped and the crocodile body is broken into two parts. Only the distal part of the tail is missing. The skull is well preserved and lacks visible fractures. The body is fully covered with an uneven layer of dark-coloured mummification balm that is notably thicker along the caudo-ventral aspects. Several fabrics, a rope, plant shoots, and a few small egg shell fragments are partially embedded in this thick layer of balm, as can easily be seen on photographs or revealed through UV illumination (**Fig 1**, 1–4). This demonstrates that a complex structure originally covered the crocodile mummy, which was removed or damaged, possibly during the exploratory unwrapping. Notably, the visible part of rope shows that it was used to arrange and stabilise the crocodile during the mummification process. The extremities of the four limbs are not visible because they are tucked in skin folds. A single incision is visible from the throat to the base of the tail (**Fig 1** and **Fig 2**). Carcass separation into two parts probably occurred quite recently, possibly during the unwrapping at the beginning of the 20[th] century, because the dividing fracture appears fresh under UV illumination and the fracture planes are not covered with balm. This fracture allows direct observation that the viscera, flesh, and bones were likely removed from the body, at least in the anterior part.

## Method

MHNL 90001850 was imaged using Propagation Phase-Contrast X-ray Synchrotron microtomography (PPC-SRμCT) with a multi-scale approach [10]. Numerous scans were performed at beamline BM05 of the European Synchrotron Radiation Facility (ESRF; Grenoble, France). The isolated head was imaged at a voxel size of 24.37 μm, whereas the body was imaged at voxel sizes of 53.19 μm and 24.37 μm. Particular details of interest were subsequently visualised

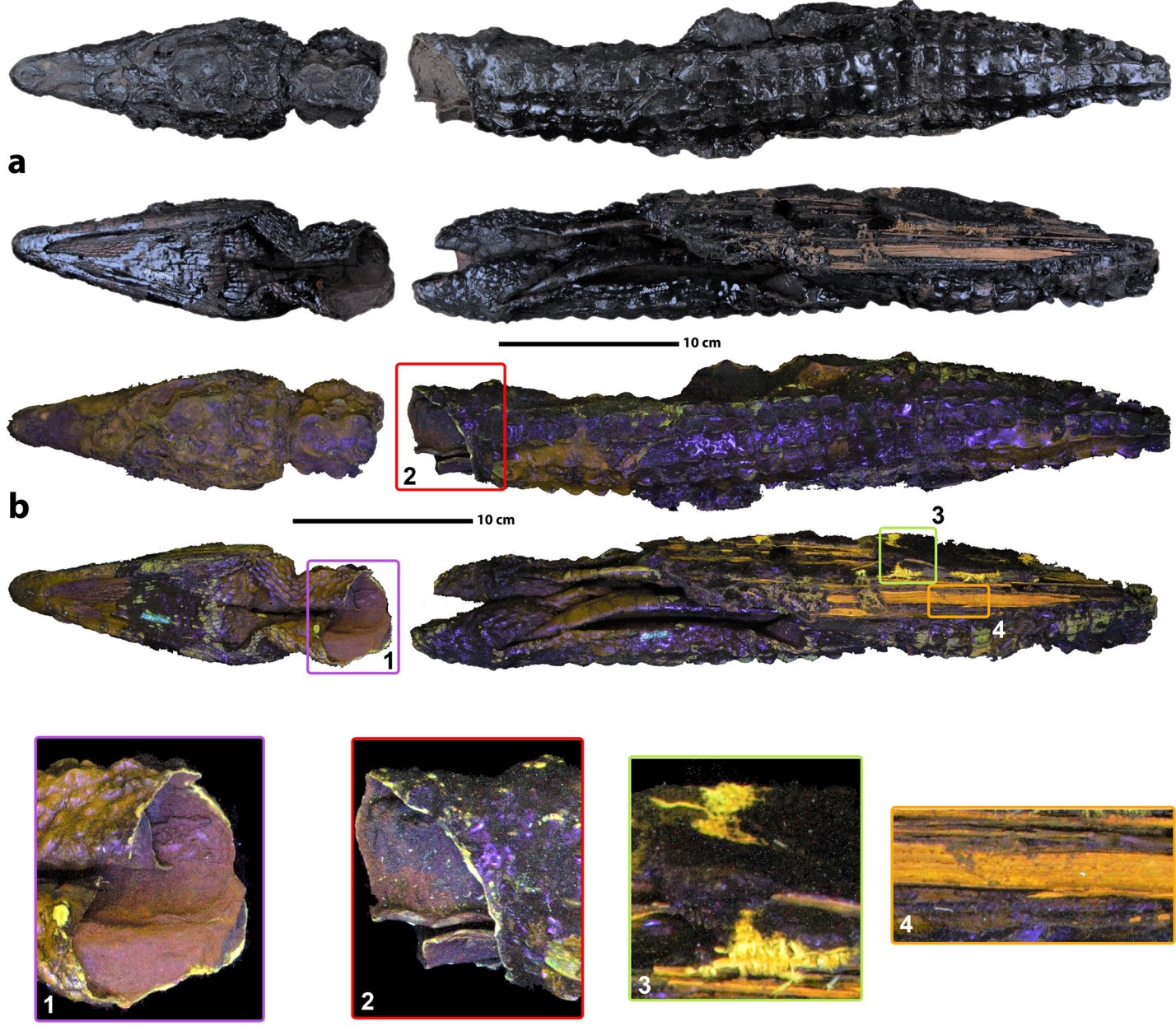

**Fig 1. Crocodile mummy MHNL 90001850. a.** Visible light picture of the specimen, **b.** UV fluorescence picture of the specimen showing in brown and dark green the parts of skin without balm, in light green the exposed skin where the body and the head are separated [1, 2], in light green the exposed osteoderms and eggshell fragments, in orange the palm-tree wood [4] and in yellow several remaining parts of the original textiles [3]. The violet parts are reflections of the lamp on shiny dark surfaces of balm.

at voxel sizes of 6.37 μm and 4.3 μm. All scanning parameters are declared in Table 1. Tomographic reconstructions were performed using a single distance phase-retrieval algorithm coupled with filtered back projection implemented in the PyHST2 software package [18–19]. To aid specific analyses, notably those on textiles, we used an algorithmic texture enhancement that was described in the supplementary information of Cau *et al.* [2017; 20]. This data processing solution was used to facilitate segmentation of the rope embedded in the balm. All 3D

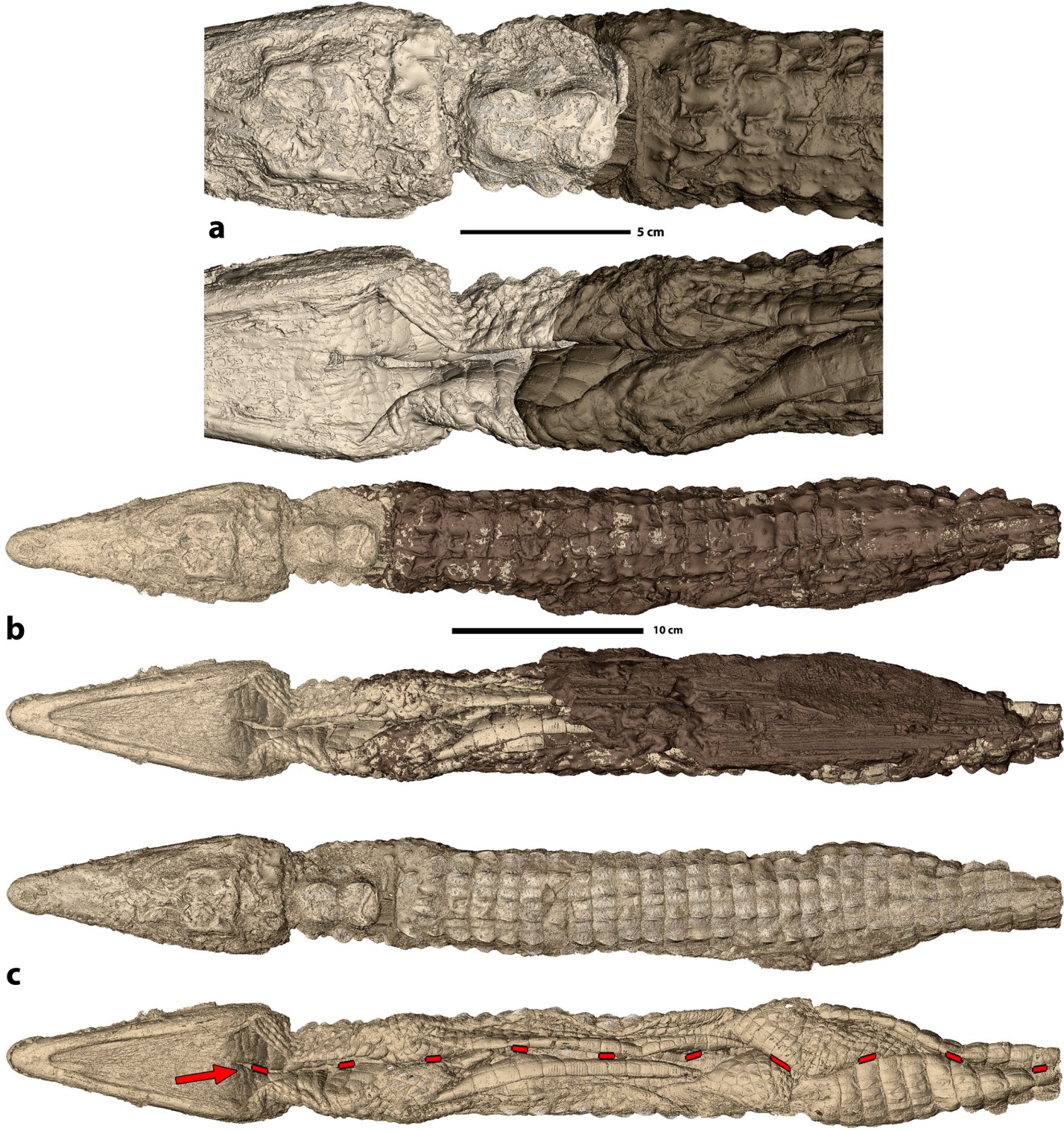

**Fig 2. 3D rendering of crocodile mummy MHNL 90001850. a.** Nearly perfect 3D match of the two preserved parts of the crocodile in dorsal (top) and ventral (bottom) view. **b.** 3D rendering of complete specimen with the thick balm in dark brown in dorsal (top) and ventral (bottom) view, **c**. 3D rendering of complete specimen without balm in dorsal (top) and ventral (bottom) view after removal of the structures overlaying the skin. Ventral view shows the incision used to prepare the body (red arrow and red broken line).

**Table 1. PPC-SRµCT data acquisition parameters.**

| Voxel size (µm) | 53.19 | 24.37 | 6.37 | 4.3 |
|---|---|---|---|---|
| Sample | Head | Head / Body | Endocast, complete insect | Plants |
| Optic | LAFIP 2 | LAFIP 2 | Hasselblad 100/100 diaphragm | Hasselblad 150/100 |
| Date | 5 May 2016 | 6 July 2016 | 18 June 2016 | 6 April 2017 |
| Average detected energy (KeV) | ~101 | ~139 | ~114 | ~120 |
| Filters (mm) | 10x Al 5, Mo 0.35 | Mo 0.3, Cu 12 | Cu 6, Mo 0.4 | Al prof 15x5, Mo 0.35 |
| Propagation distance (mm) | 4000 | 4000 | 2600 | 2500 |
| Sensor | FreLoN 2K14 | PCO Edge 5.5 | PCO Edge 5.5 | PCO Edge 4.2 CLHS |
| Scintillator | LuAg 2000 | LuAg 2000 | GGG 500 meniscus | LuAg 200 meniscus |
| Projection number | 2799 | 4000 | 3999, 6000 | 5000 |
| Scan geometry | 360 degrees scans, vertical scan series with 2.5 mm translation | Half Acquisition 600 pixels offset. Vertical scan series with 1.5 mm translation | Local tomography, Half Acquisition 1000 pixels offset, Vertical scan series with 1.6 mm translation | Vertical scan series with 4 mm translation |
| Exposure time (s) | 0.025 | 0.02 | 0.035 | 5x0.01s |
| Number of scan | 85 | 268 | 35 / 64 | 49/77 |
| Reconstruction mode | Single distance phase retrieval (Paganin 2002), vertical concatenation, ring artefacts correction, 16 bits conversion in jpeg2000 format | Single distance phase retrieval (Paganin 2002), vertical concatenation, ring artefacts correction, 16 bits conversion in jpeg2000 format texture recoding | Single distance phase retrieval (Paganin 2002), vertical concatenation, ring artefacts correction, 16 bits conversion in jpeg2000 format | Single distance phase retrieval (Paganin 2002), vertical concatenation, ring artefacts correction, 16 bits conversion in jpeg2000 format |

renderings and segmentations were performed using VGStudioMax 3.1 and 3.2 (Volume Graphics, Heidelberg, Germany).

All tomographic data and segmentation files used for this study are available through the open access tomographic database of the ESRF (http://paleo.esrf.eu).

## Results

### 1. Description from synchrotron X-ray imaging

PPC-SRµCT allowed for the evaluation of various anatomical details preserved in the crocodile mummy. The anterior and posterior parts of the specimen were first retrofitted in 3D. This revealed a nearly perfect fit of the two pieces (**Fig 2A**), which confirmed that they indeed represent two parts of the same specimen and corroborated that no substantial intermediate portion is missing. As such, this observation indicates that the separation between the head and the postcranium is likely quite recent. Once matched, the body measures 55 cm in preserved length (**Fig 2B**). The skin envelope is virtually empty. All organs and most of the bones have been removed through to the single incision that runs along the ventral part of the crocodile from the throat to the tail. This incision is clearly resolved on the 3D renderings after virtual removal of the covering structures (**Fig 2C**). The bones still present in the body, i.e. the skull, the phalanges, and the osteoderms (**Fig 3A**), all represent elements that could not be removed without damaging the skin further.

3D segmentation revealed cut marks on the proximal aspects of nearly all remaining limb bones (red arrows in **Fig 3B and 3C**). Cut marks lacking striations are present on the anterior left metapodials I, II, III, IV (**Fig 3B2**), anterior right metapodials II, III, IV (**Fig 3B1**), posterior left metapodial I (**Fig 3B4**), posterior right metapodials I, IV (**Fig 3B3**), and several posterior left and right tarsals (**Fig 3B3 and 3B4**). These cut marks are associated with bones

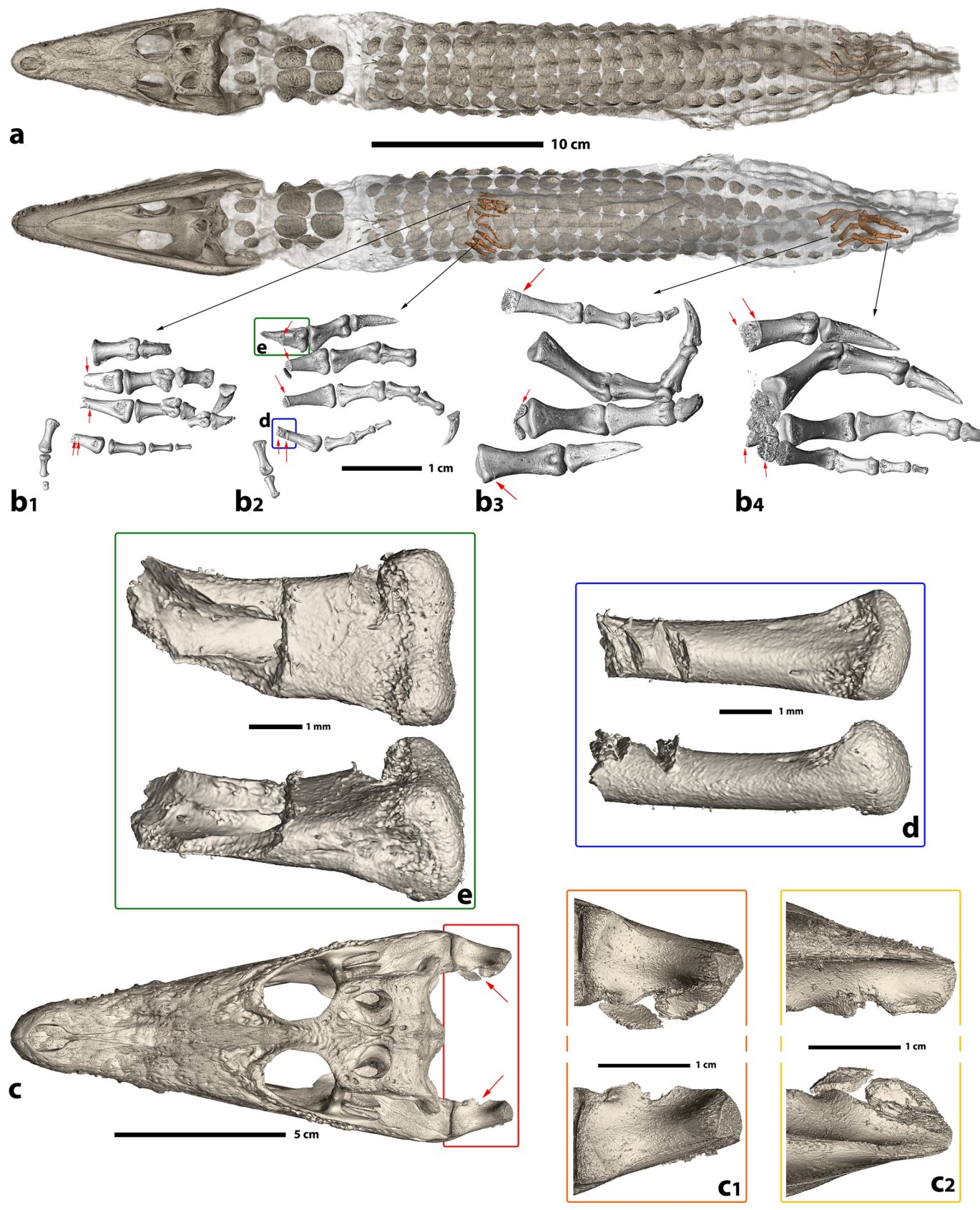

**Fig 3. 3D rendering of the preserved bones inside crocodile mummy MHNL 90001850. a.** Locations of preserved bones (skull, phalanges, parts of metatarsal bones, and osteoderms) inside the mummy (here rendered as semi-transparent surface), **b.** details of the limb bones still present in the body: b1. right manus in dorsal view, b2. left manus in palmar view, b3. right pes in palmar view, b4. left pes in palmar view; red arrows indicate cut marks, **c.** locations of cut marks in the mandible, **c1.** detail in dorsal view, **c2.** detail in ventral view, **d.** and **e.** cut marks on the metacarpal IV (**d.**) and digit I (**e.**) of the left manus.

fractures. We also observed cut marks on the retroarticular processes of the mandible (**Fig 3C**, red arrows) [21].

## 2. Vegetal infillings of the emptied limbs

Beside inference of carcass preparation protocols for mummification, PPC-SRμCT data also allowed for recognition of the various foreign elements that were added to the body during the mummification process.

The emptied limbs have been stuffed with a variety of plants, most of them grasses (**Fig 4**). Although conclusive identification of these plants was outside of the scope of this study, several specimens are well preserved (**Fig 4C** and **4D**) and could therefore probably be determined down to generic or even specific level. Conclusive recognition in future studies would offer additional information on the surroundings and context of this peculiar mummification practice.

## 3. External additions

Some remains of the original linen bandages are still visible on the surface of the mummy (visible in yellow on the ventral side under UV light in **Figs 1** and **3**), or embedded in the balm (**Fig 5C**). In addition, a distinct textile layer can be recognised separating two different layers of balm (**Fig 5C** and **5D**). The innermost balm layer contains numerous dense inclusions whereas the top layer is nearly devoid of inclusions (**Fig 5D**).

Fragments of vegetal shoots were encountered still embedded in the balm and on the surface of the balm (**Figs 1** and **4**, visible in orange under UV illumination, **Fig 5E**). Their transverse microstructures, visible on the tomographic data, suggest that they represent leafstalks from the date palm (*Phoenix dactylifera*) [22].

The balm present on the caudoventral part of the crocodile has a maximum thickness of 21 mm (**Fig 4B**).

The rope used to keep the crocodile skin in the desired arrangement is not only present on the surface, but also resides partially embedded in the balm (**Fig 5B**). Constrictions marks on the skin at certain locations indicate the original trajectory of this rope around the animal (**Fig 5B**) before it was mostly removed during the unwrapping process, which probably occurred during previous studies.

## 4. Insects

Within the mummy, insect remains are exclusively present in skull cavities of the crocodile (**Fig 6**). Among these is a complete adult coleopteran specimen that has been tentatively identified as the hide or leather beetle (*Dermestes maculatus*) [23] (**Fig 6D**, S1 Movie). It is preserved suspended from the cartilaginous interorbital septum of the crocodile, indicating that it resided there before the carcass was dehydrated during mummification [24]. Its internal anatomy, although strongly desiccated, is preserved (S1 Movie). Other insect remains include the other developmental stages of *Dermestes* preserved as desiccated eggs and larvae (**Fig 6A** and **6E**). Most of the eggs are hatched and measure 3.2 mm and 0.9 mm in average length and diameter, respectively. All insect remains were encountered located in cranial cavities (**Fig 6A**), with only few small fragments additionally observed in mandibular cavities. Notably, the

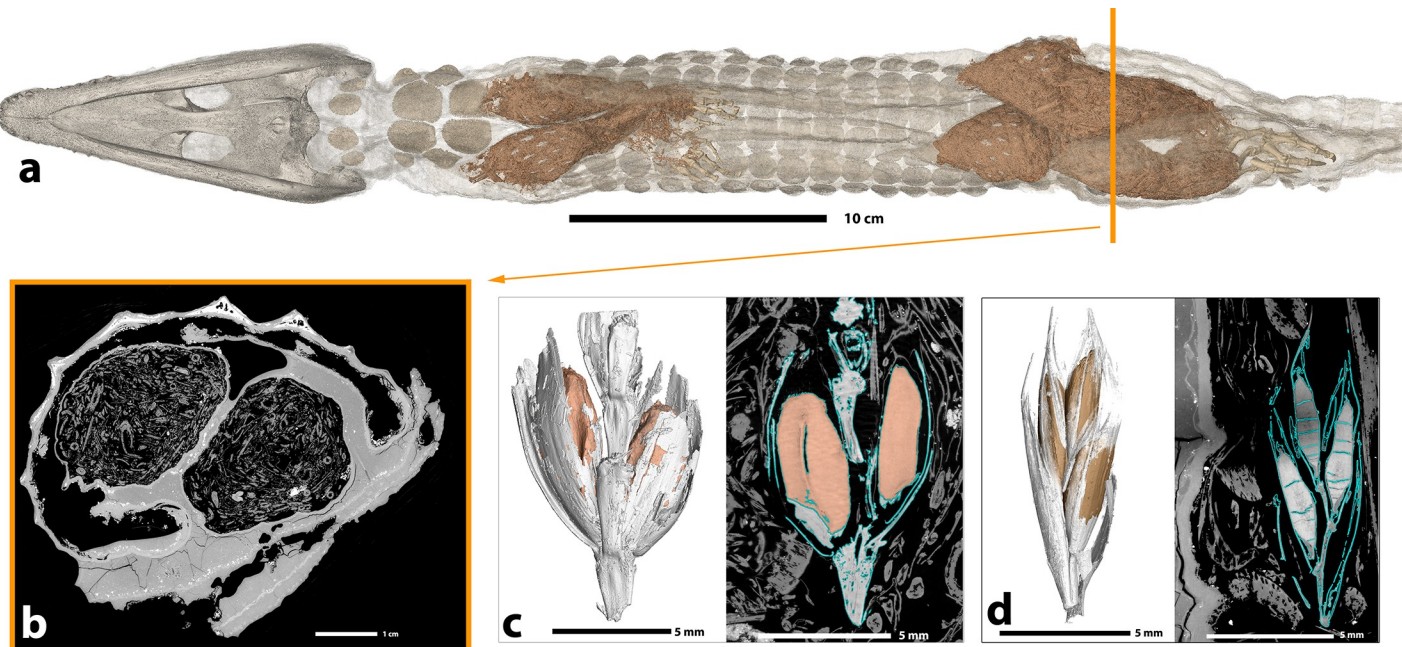

**Fig 4. Details of the filling of the crocodile mummy MHNL 90001850. a.** 3D rendering of the crocodile highlighting the filling, **b.** tomographic slice showing vegetal limb stuffing and thickness of the balm, **c.** and **d.** 3D renderings (left) and tomographic slices (right) of two of the well-preserved ears of grass within the filling.

balm layer or remaining soft tissues in the head of the mummy do not preserve any indications for the presence of insect entrance or escape burrows. Crucially, insect traces are completely absent in the oral cavity and the skin of the crocodile.

## 5. Mineral inclusions in the first layer of balm

Although the balm in the mummy has not yet been chemically analysed, information inferable from tomographic data provide certain clues on the material properties of the balm. Despite the tomographic data having been acquired using a polychromatic beam, simulation of the detected X-spectrum shows a negligible effect of beam hardening. This enables relatively accurate measurements of the linear attenuation coefficients ($\mu$, in cm$^{-1}$) of obscured media, as if the scans would have been performed using a monochromatic beam. A first estimation of effective energy based on spectrum simulation (139 keV) was refined by calibrating the measured absorption of cortical bones ($\mu = 0.274$ cm$^{-1}$) in the mummy with the measured mineral (hydroxyapatite) density of a modern clean and dehydrated three-year-old *Crocodylus niloticus* humerus (1.8 g/cm$^3$). This calibration indicated an effective energy of 146.1 keV. This higher estimation can be attributed to the diffusion of light in the crystalline scintillator of the detector. Since effective energy was calibrated against a known sample, this deviation is not problematic for subsequent measurements. We compared the range of measured $\mu$ values for mineral inclusions (average $\mu = 0.297$ cm$^{-1}$, maximum $\mu = 0.334$ cm$^{-1}$) against calculated theoretical $\mu$ values of the main constituting minerals of natron salt used for desiccation practices in ancient Egypt (i.e. $Na_2CO_3.10(H_2O)$, $NaHCO_3$, $NaCl$, $Na_2SO_4.10(H_2O)$, and $Na_2CO_3.NaHCO_3.2H_2O$ [25–28]). For the X-ray energy range used, we found that the encountered mineral inclusions exhibit $\mu$ values agreeing with those of the main constituting minerals of natron (**Table 2**).

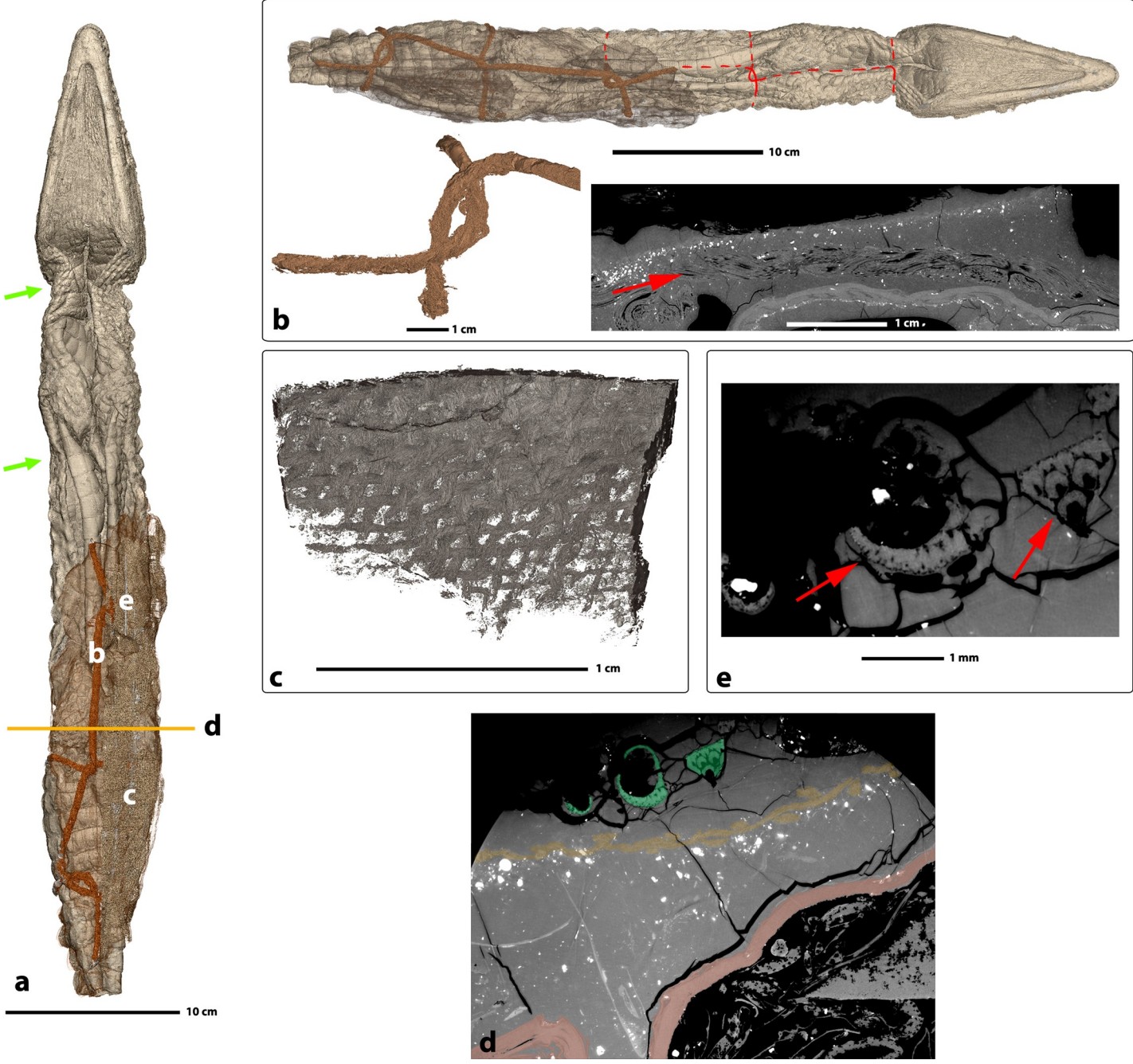

**Fig 5. Details of the inclusions present inside the crocodile mummy. a.** general 3D rendering of the inclusions and their locations, **b.** 3D rendering and tomographic slice of the rope embedded in the balm (dark brown) and putative location of the missing parts of the rope inferred from the constrictions of the skin (green arrows), **c.** 3D rendering of the fabrics present inside the balm, **d.** tomographic slide showing the denser mineral inclusions (white spots in balm) below a layer of fabric (highlighted in pastel yellow) and the date palm leaves in green, **e.** tomographic slide of the date palm (*Phoenix dactylifera*) leaves rachis.

# Discussion

## 1. Experimental archaeology

To better understand the unusual pre-mummification preparation protocol recorded in MHNL 90001850, we performed an experimental test. We used a *Caiman crocodilus* individual

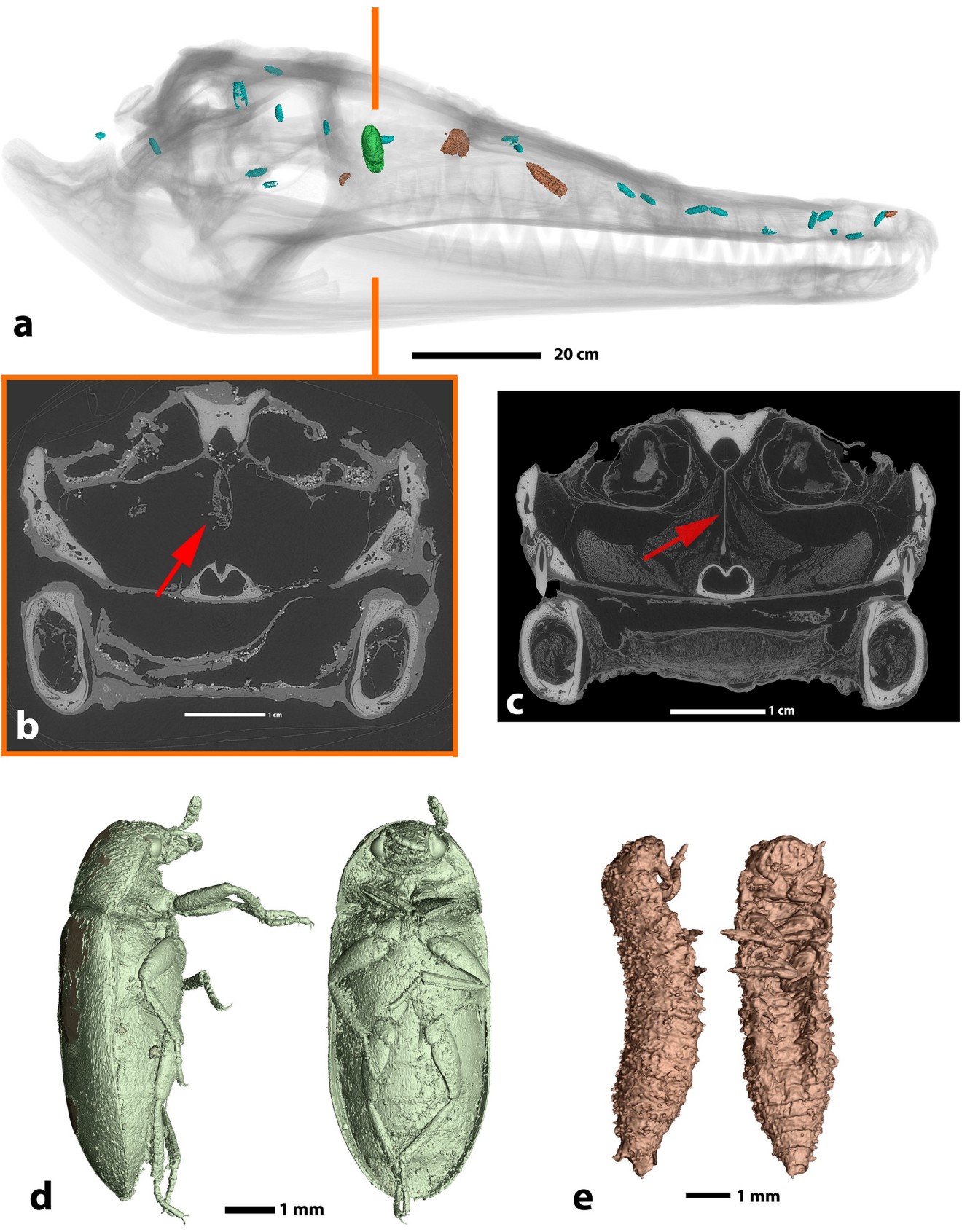

**Fig 6. Insect remains encountered in the skull of crocodile mummy MHNL 90001850. a.** locations of the remains: the only adult specimen appears in green, the larvae in brown, the eggs in light blue, **b.** transverse tomographic slice of the skull through the orbits illustrating the location of the adult specimen (red arrow) suspended upside down from remains of the interorbital septum, **c.** location of the interorbital septum in a modern desiccated crocodilian (red arrow), **d.** 3D rendering of the adult specimen of *Dermestes maculatus*, **e.** 3D rendering of a larva attributable to the same taxon.

that anatomically matches the Nile crocodile with the notable additional presence of ventral osteoderms, which are absent in the Nile crocodile. This specimen died unexpectedly at the "Reptilarium des Landes" in Labenne, France, and was preserved frozen for a several months. First, we made an incision from the throat down to the base of the tail. To negotiate the ventral osteoderms, which grant much higher ventral skin rigidity than in Nile crocodiles, we made two more incisions. These were placed perpendicularly to the longitudinal cut, with the first one positioned a few centimetres posteriorly to the anterior limbs and the second one a few centimetres anteriorly to the posterior limbs (**Fig 7A**). Using a regular stainless steel knife, we removed all organs, muscles, and accessible bones (**Fig 7B**) in one mass. This was achieved by reverting the skin (i.e. turning it inside out) after having separated the skull from the vertebral column by inserting the blade between the atlas and the occipital condyle. The limbs were prepared in a similar fashion by turning the skin inside out to expose the muscles and bones as much as possible. It was possible to reach the metapodal level in the distal limbs. As this specimen was intended for an osteological preparation, metapodal elements were not broken as in the Egyptian mummy, but carefully separated at the level of their proximal joints. The skin of the tail was separated from muscles and bones up to one third of the length of the tail, but this could have been continued further down. When all the remaining contacts between the skin and the rest of the body were severed, the skin was placed back in its original orientation. The emptied limbs were stuffed with locally sourced plant material (dried grasses) (**Fig 7C**). The specimen was secured in its ultimate pose using a rope (**Fig 7D** and **7E**). Performing the described preparation on a small Nile crocodile would have certainly been less demanding than on a circa 150-cm-long caiman, as the skin would have been thinner and more flexible.

## 2. Shape and aspect of the mummy

As also seen with the caiman experiment, filling the limbs with dried plants and tucking them inside the body outline appears to offer a very efficient solution to shaping a crocodile skin and grant it body contours that look almost natural in dorsal view (**Fig 7E**). This method does not require sewing, which would be difficult and time consuming to perform due to the rigidity of crocodile skin. The associated bandages, the twigs, and the posture in which the body was arranged suggest that this crocodile was elaborately prepared with textile to give it the appearance of a mummified crocodile.

## 3. Cut marks

As MHNL 90001850 is nearly empty and presents clear cut marks on the bones that do remain, it is evident that the body was purposely cut and extensively cleaned by Ancient Egyptians.

**Table 2. Details and properties of the main constituting minerals of natron.**

| mineral composition | density (g/cm$^3$) | μ at 146.1 keV (cm$^{-1}$) | mineral name |
|---|---|---|---|
| $Na_2CO_3.(0\ to10)(H_2O)$ | 2.54–1.44 | 0.345–0.21 | sodium carbonate (from anhydrate to decahydrate) |
| $NaHCO_3$ | 2.21 | 0.3050 | sodium bicarbonate, nahcolite |
| NaCl | 2.17 | 0.3130 | sodium chloride |
| $Na_2SO_4.(0\ to10)(H_2O)$ | 2.664–1.464 | 0.373–0.215 | sodium sulphate (from anhydrate to decahydrate) |
| $Na_2CO_3.NaHCO_3.2H_2O$ | 2.11–2.17 | 0.298 | trona, trisodium hydrogendicarbonate dihydrate |

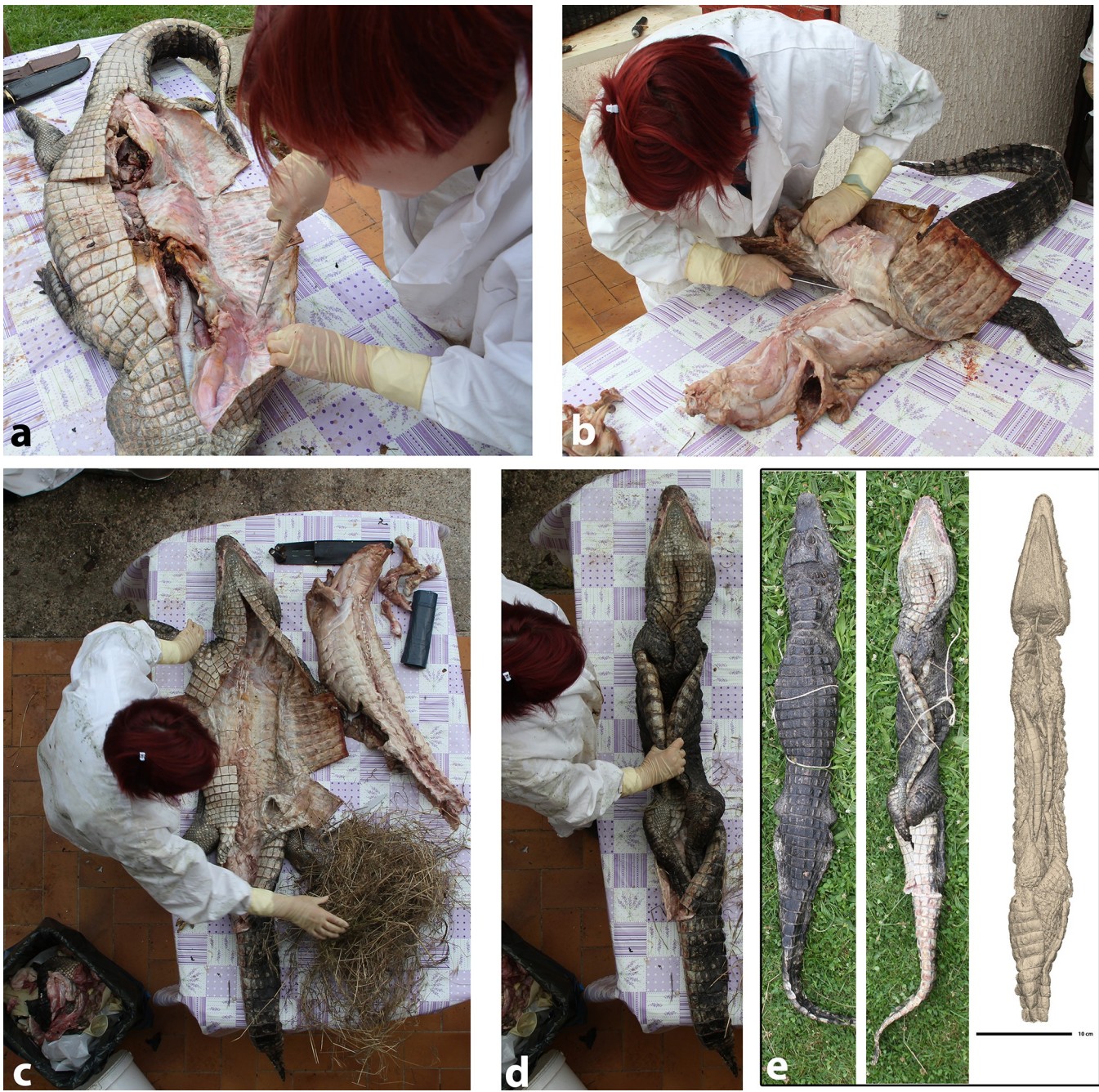

**Fig 7. Experimental preparation of a caiman cadaver. a.** The skin is separated from all other tissues, **b.** After separating the skull from the neck, the skin is turned inside out to facilitate its separation over the dorsum and enable evacuation of the limbs, **c.** The limbs are stuffed with plants (grasses) **d.** The limbs are tucked inside the skin and the body is "shaped", **e.** Comparison between the result of the experiment and the 3D rendering of the genuine Egyptian mummy.

Geometry and location of the cut marks can be explained by a single move repeated several times when the bones were not separated after the first cut. The more distal cut marks appear as cracks, which may indicate a percussive action rather than a slicing movement [29]. Multiple cut marks in a single bone indicate that the manual digits were severed by repeated cuts at the level to which the metapodal bones could be reached when the skin was reversed as much

as possible (there are no cut marks on the skin at the corresponding levels). The smoothness of the cut marks and their V-shaped profiles appear more consistent with the use of a metallic tool, as opposed to rougher marks that would be expected following employment of lithic blades [30, 31]. However, scans at higher resolutions would be required to offer more certainty regarding the type of tools used on this specimen. Cut marks on the skull are different in that they appear more like break marks than cut marks. They probably reflect multiple blows of the tool used to separate the skull from the neck to enable reversal of the skin.

## 4. Insects

*Dermestes* are commonly found in mummies. They usually accessed the carcass during or after mummification [32]; sometimes even very recently in cases of non-adapted storage conditions. However, in the case of the mummy MHNL 90001850, we propose that *Dermestes* colonised the crocodile carcass before mummification but following an episode of advanced decay. [33] Remains of these insects, including their eggs and larvae, are exclusively found in the cranial region (i.e. cranium and mandibular cavities), but are absent elsewhere, notably including the oral cavity. In case of infestation during or after the mummification process, the insects would be present throughout most of the specimen, and certainly in the oral cavity, rather than being restricted to exclusively isolated cranial cavities. In addition, the adult *Dermestes* individual was found sticking to a cartilaginous membrane in the skull (the interorbital septum), although such a structure would have lost adhesive properties after completion of the dehydration process. Finally, the insect body itself has clearly experienced mummification through rapid dehydration alongside the crocodile cadaver. We therefore interpret that the decaying crocodile remains represented a cadaver of several days or even weeks old when it was used to manufacture the mummy. The complete body was emptied, thereby removing all body parts that could continue to decay, with exception of those skeletal elements that were too closely associated with the skin (i.e. the skull, mandible, digits, and probably the unpreserved distal caudal vertebrae). As we find insect remains only in the cranial cavities but not in the oral cavity, we can deduce that the body was thoroughly cleaned after removal of all muscles, internal organs, and skeletal elements, except where cleaning was not possible; i.e. the inaccessible cranial cavities. The cadaver was prepared into a cleaned skin with only the skull and digits still in place, as they could not be removed without further damaging the skin. Since there was no further infestation by insects after cleaning procedure and we did not observe any entrance or escape tunnels through the balm or the remaining soft tissues, we deduce that the insects must have perished during the cleaning process or during the following desiccation in natron that resulted into the inadvertent mummification of the adult *Dermestes* individual.

## 5. Manufacturing stages

Through the various inclusions encountered within the balm layer, we can propose a protocol explaining how this particular mummy was manufactured. Most organs and tissues were removed from the crocodile body, the skin was washed (**Fig 8A**), and the limbs were stuffed with plants (**Fig 8B**) before the carcass was fixated with rope (**Fig 8C**). As we observe natron salt in direct contact with the skin (**Fig 8D**), we conclude that a first desiccation procedure was carried out that involved placing the corpse in solid natron. Then, a first layer of balm (**Fig 8E**) was applied and covered with textile (**Fig 8G**). This balm contained natron (or a mineral mixture with similar X-ray absorbing properties), possibly to enhance desiccation of the specimen and avoid further decay. The distribution of natron inclusions throughout the entire balm layer, rather than mostly in direct contact with the skin, is more compatible with the intended application of a balm-natron mixture than with contamination of the balm following

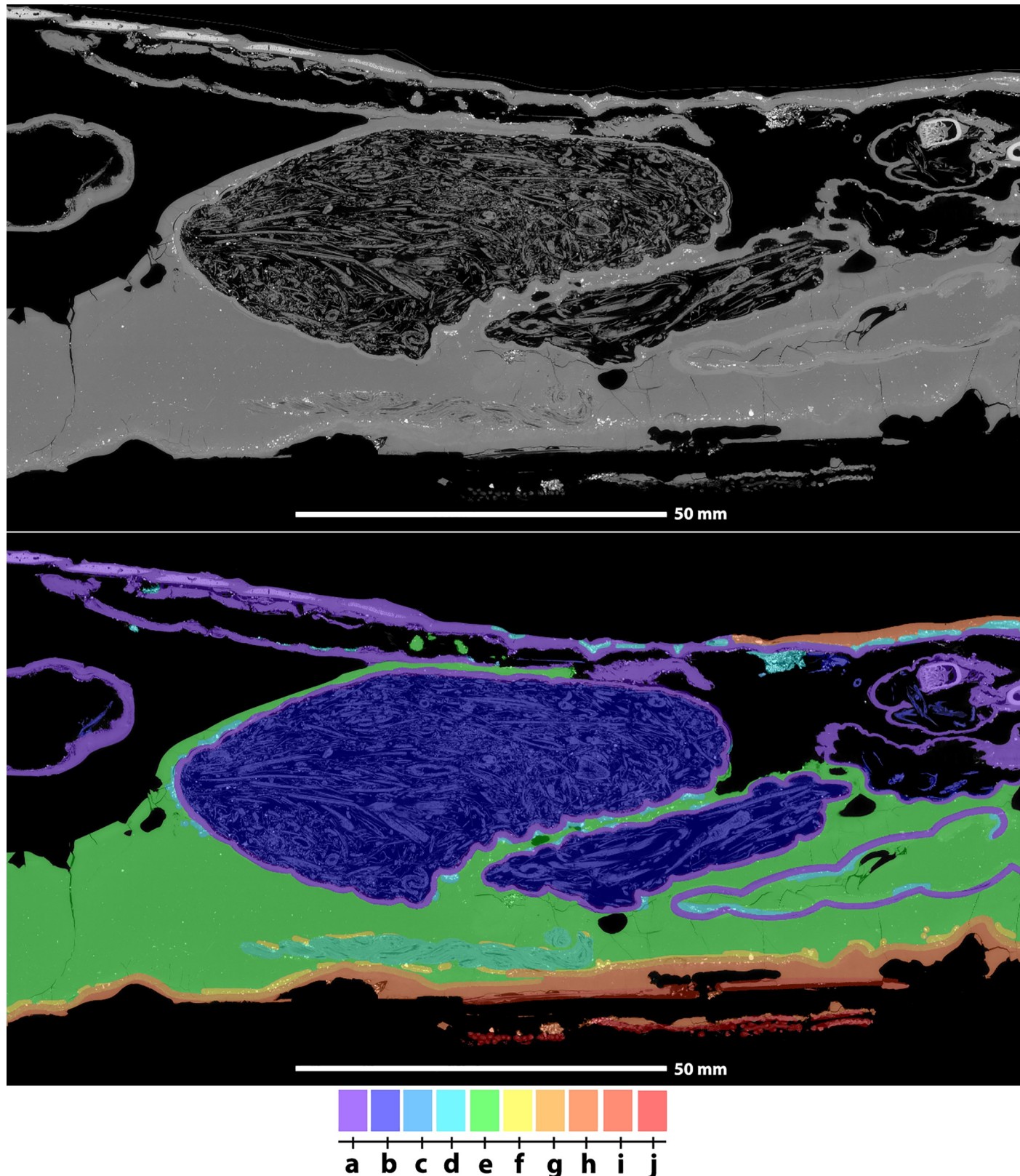

**Fig 8. Tomographic slides from posterior limb region, in sagittal view, illustrating the different steps in the manufacturing process of this particular crocodile mummy.** The colour gradient corresponds to the successive stages of the manufacturing protocol for this mummy (From **a.** to **j.**, see main text for stage description).

application on a specimen that was not cleaned of all natron used for initial desiccation. The crocodile carcass was treated while being positioned its ventral side. Higher natron concentrations are present in the lower regions (i.e. around the ventral aspects of the crocodile) of the first balm cover (**Fig 8F**), which we interpret to reflect natron settling during solidification of this balm. Following this stage, a second layer of balm (**Fig 8H**), not enriched with natron, was applied on mainly the ventral aspects of the crocodile cadaver. Finally, the embalming priests added parts of palm leaves (**Fig 8I**) and textiles (**Fig 8J**) to achieve more rigidity and to achieve the intended shape. This approach is often observed in small crocodile mummies and occasionally in larger specimens [12, 34]. The mummy was finally wrapped in various bandages to achieve the desired appearance.

## Conclusions

Votive animal mummies were not always created from a single well-preserved animal cadaver. Cases have been described were mummies contained decayed animal body parts. Nevertheless, the peculiar preparation process of the MHNL 90001850 crocodile mummy represents a very efficient approach to manufacturing a mummy with a cadaver in an advanced stage of decomposition. By removing the decaying parts infested by necrophagous insects as much as possible, Ancient Egyptians reduced the risks of continued decay after finalisation of the mummification process, resulting *de facto* in a preparation quite similar to taxidermy. This hypothesis explains all our observations, but we can evidently not rule out that this manufacturing process was (also) linked to a specific religious practice or idea.

The lack of archaeological provenance and context for this specimen renders it very difficult to conclusively place it in a chrono-cultural context. Richardin *et al*. published radiocarbon dates for several samples from the collection of the Musée des Confluences [7]. All crocodiles dated in that study brought results between 2255±30 BP and 1845±30 BP (i.e. from the end of the Late period to the middle of the Roman period), which corresponds to the period when the animal mummy industry in Ancient Egypt was active at a very large scale. Within this context, being able to use any available cadaver would have been beneficial to secure the supply of material for the production of animal mummies. As demonstrated by our experiment on the modern caiman, this protocol would have offered a quite simple and efficient solution to make an opportunistically discovered decaying cadaver available for mummification. The future availability of more reliable temporal information as well as the study of similarly prepared mummies (if any) with known context should grant more comparative insight into this unusual mummification protocol.

## Supporting information

**S1 Movie.**
(MP4)

## Acknowledgments

We would like to thank the "Musée des Confluences", and in particular Didier Berthet, for having given us the opportunity to work on MHNL 90001850. We thank "La Ferme au Crocodiles", especially Samuel Martin, for their support and assistance during initial phases of the project. We thank Laura Jarnias who assisted us during data segmentation. We are grateful to Thierry Loeb and the Reptilarium des Landes for allowing us to perform the actuo-archaeological experiment on the specimen of *Caiman crocodilus*. We are grateful to the ESRF, and the BM05 team to have granted the in-house beamtime used to perform multiple scans of this very

special mummy. We also acknowledge David Lefèvre and Frédéric Servajean for their support to the MAHES project. We are very grateful to Dennis Voeten who dramatically improved the English quality of the present text. Finally, we are grateful to the editor, Jana Jones and the anonymous reviewer for their constructive comments and suggestions. All the tomographic data and segmentation files used for the present study are available through the open access tomographic database of the ESRF (http://paleo.esrf.eu).

## Author Contributions

**Conceptualization:** Stéphanie M. Porcier, Paul Tafforeau.

**Data curation:** Paul Tafforeau.

**Formal analysis:** Camille Berruyer.

**Funding acquisition:** Stéphanie M. Porcier, Paul Tafforeau.

**Investigation:** Camille Berruyer.

**Methodology:** Camille Berruyer, Paul Tafforeau.

**Project administration:** Stéphanie M. Porcier, Paul Tafforeau.

**Supervision:** Stéphanie M. Porcier, Paul Tafforeau.

**Validation:** Stéphanie M. Porcier, Paul Tafforeau.

**Visualization:** Camille Berruyer.

**Writing – original draft:** Camille Berruyer.

**Writing – review & editing:** Stéphanie M. Porcier, Paul Tafforeau.

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
