## [Decision Letter · Decision Letter 0]

16 Dec 2019

PONE-D-19-24542

Synchrotron “virtual archaeozoology” reveals how Ancient Egyptians prepared a decaying crocodile cadaver for mummification

PLOS ONE

Dear Camille Berruyer,

Thank you for submitting your manuscript to PLOS ONE. After careful consideration, we feel that it has merit but does not fully meet PLOS ONE’s publication criteria as it currently stands. Therefore, we invite you to submit a revised version of the manuscript that addresses the points raised during the review process.

We would appreciate receiving your revised manuscript by Jan 27 2020 11:59PM. To enhance the reproducibility of your results, we recommend that if applicable you deposit your laboratory protocols in protocols.io, where a protocol can be assigned its own identifier (DOI) such that it can be cited independently in the future. For instructions see: http://journals.plos.org/plosone/s/submission-guidelines#loc-laboratory-protocols

We look forward to receiving your revised manuscript.

Kind regards,

William Oki Wong, Ph.D.

Academic Editor

PLOS ONE

Journal Requirements:

**When submitting your revision, we need you to address these additional requirements:**

**Please ensure that your manuscript meets PLOS ONE's style requirements, including those for file naming. The PLOS ONE style templates can be found at http://www.plosone.org/attachments/PLOSOne_formatting_sample_main_body.pdf and http://www.plosone.org/attachments/PLOSOne_formatting_sample_title_authors_affiliations.pdf**

2. We note that Figure 7 includes an image of a participant in the study. 

Reviewers' comments:

Reviewer's Responses to Questions

**Comments to the Author**

1. Is the manuscript technically sound, and do the data support the conclusions?

Reviewer #1: Yes

Reviewer #2: Partly

2. Has the statistical analysis been performed appropriately and rigorously? 

Reviewer #1: Yes

Reviewer #2: N/A

3. Have the authors made all data underlying the findings in their manuscript fully available?

Reviewer #1: Yes

Reviewer #2: Yes

4. Is the manuscript presented in an intelligible fashion and written in standard English?

Reviewer #1: No

Reviewer #2: No

5. Review Comments to the Author

Reviewer #1: Please see my Review uploaded as attachment, many thanks.

Reviewer #2: PONE crocodile cadaver mummy

This a very interesting article, but there are a few issues. Firstly the language needs revision by a native speaker. Then, other points: The type of preservation of this animal is very interesting. The experimental work is a great plus. However, the arguments for the fact that a semi-decomposed cadaver was used are not compelling. Perhaps the authors would revise the English (perhaps some of the arguments loose their clarity due to language issues), and also revisit their conclusions, maybe presenting the cadaver theory as a possibility, but not the only one. Also, the authors should explore why the ancient Egyptians would have removed all of the bones—even if they were using a slightly putrefied cadaver, one would expect the flesh and internal organs to be removed, not the bones. Speculation would be fine.

Specifis (including language issues)

-Do not capitalise Ancient (as in Ancient Egyptian should be ancient Egyptian)

l. 29 granted completely continuous access is awkward phrasing

l. 40-46 somewhat unclear in meaning, mainly due to language issues

l. 50 better if: several individuals representing the same species were to read several individuals of the same species

l. 53 Instead of mummification preparation protocol the authors should say mummification, or alter the phrasing. Also, awkward phrasing: involving cessation of putrefaction has been brought forward.

l. 61 should say provenanced. Would be useful to have the dimensions of the animal (skull and post-cranial)

l. 64 not driven by…. Choose another word

l. 66 unclear: alathough (17)) do mention both Esna…

l. 67-68 no proof that bandaging provides chronological data or provenance data

l. 68 should read on the period of the..

l. 81 maybe better to say dark coloured mummification materials (balm) and then refer to it as balm thereafter

l. 82 Unclear: The limbs are hidden in skin folds

l. 83 Provide more precise details of the location of the incisions. Maybe on an image? Also, break, not brake, and indeed, the whole sentence following from ‘brake’ needs to be revisied for clarity (clearly a problem with language).

l. 85-87 Unclear: Fractures into the balm layer and the scattered remains of textile, rope, 85 plants shoots, and other isolated elements indicate that certain parts of the original external layers were removed from the mummy.

l. 110 Not sure about anatomical properties

l. 136: Thus, only osteoderms, phalanges, skull preserved, almost all with cut marks? But then l. 137 mentions metapodia. Maybe authors should provide a list of all the bones that appear in fig. 3

l. 146 maybe vegetal or a variety of plant matter rather than floral? It is only a quibble.

l. 157 Abrupt introduction of rope…

l. 168 Awkward English

l. 181 English issues : dorsal aspects is stick to a remaining part of the cartilaginous

l. 278 When mummies are being produced dermestids can enter the body, as seems to be the case with this animal. Sometimes, even when the animal is covered with natron, these beetles can enter the body and carry out their entire life cycle during the course of desiccation. See: Ikram, S. 2015. ‘Experimental Archaeology: From Meadow to Em-baa-lming Table’, in C. Graves-Brown (ed.) Egyptology in the Present: Experiential and Experimental Methods in Archaeology, pp. 53-74. Swansea: The Classical Press of Wales. The authors have to argue more convincingly that the crocodile had been dead for some time when it was eviscerated and basically turned into a taxidermied crocodile in term of having most of the bones, flesh and organs removed and being stuffed with grassed (but after being traditionally desiccated using natron, and then covered with the conventional balm?).

l. 290 the skin was cleaned—maybe the authors could suggest with what?

l. 293 Possible that the natron was not well cleaned off the body, which is why it stayed on, rather than having the balm mixed with natron. Perhaps analyses of the balm in these areas would clarify the situation. SEM?

l. 300 Palm leaves, particularly the ribs, and textiles are traditionally used to make crocodiles (and sometimes humans) more rigid and stackable. Generally this is more common in immature crocodiles, but is also known in mature animals.

6. PLOS authors have the option to publish the peer review history of their article (what does this mean?). If published, this will include your full peer review and any attached files.

Reviewer #1: Yes: Jana Jones PhD

Macquarie University

Sydney Australia 2106

Reviewer #2: No

---

## [Author Response · Author response to Decision Letter 0]

30 Jan 2020

Response to reviewer

We are very grateful to the editor and to the reviewers for their constructive comments and corrections. As the English level was clearly a problem, we asked to one of our colleague to revise the whole text to improve its quality. As a result, the version with track changes will show a huge amount of modifications, but the vast majority are linked to this English improvement. For all the other questions and comments not directly linked to the general English level, we reply point-by-point hereafter to the best of our capacities.

Reviewer #1

-l. 42 Delete ‘content’, substitute ‘composition’. 

→ Done, please see in the modified text

-l. 62 Delete space after ‘information’ and before the full stop. 

→ Done, please see in the modified text

-l. 66 insert ‘Lortet and Gaillard (17.)’ Just the reference number is not sufficient. 

→ Done, please see in the modified text

-l. 67 ‘However, as all the textiles have been removed, it is now impossible to reliably derive the origin …. through manufacturing techniques’. It would not have been possible to ‘reliably derive the origin’ unless there was evidence of an especially unusual and datable weaving or spinning technique. I would suggest: ‘it is now impossible to gain insights into the origin of this mummy through any distinctive manufacturing techniques’. 

→ Done, please see in the modified text

-l. 68 Delete ‘elucidate on’, insert ‘reveal’. 

→ Done, please see in the modified text

-l. 77 Delete ‘relative to’, substitute ‘in’

→ Done, please see in the modified text 

-l. 77 Delete ‘hence we propose hereafter …’ insert (new sentence) ‘Hence we propose the following updated description’. 

→ Done, please see in the modified text

-l. 78 Delete ‘already’, insert ‘have’. 

→ Done, please see in the modified text

-l. 81 ‘plants’. Delete ‘s’, = singular ‘plant shoots’. 

→ Done, please see in the modified text

-l. 82 Delete ‘stick’. 

→ Done, please see in the modified text

-l. 83 Delete ‘brake’, insert ‘break’. Misspelling. 

→ Done, please see in the modified text

-l. 84 Delete ‘indicating’, insert ‘indicate’. Delete ‘corps’, insert ‘corpse’, or better still, ‘body’. 

→ Done, please see in the modified text

-L. 85 Delete ‘into’, insert ‘in’ the balm layer’. 

→ Done, please see in the modified text

-l. 169 Insert space after ‘surface’ and before ‘Figure 1’. 

→ Done, please see in the modified text

-l. 181 ‘Its dorsal aspects is stick’. Delete s = singular ‘dorsal aspect. Delete ‘is stick’, insert ‘adheres’, i.e. ‘Its dorsal aspect adheres’. 

→ Done, please see in the modified text

-ll. 196-211 NBB The composition of the ‘balm’ has not undergone chemical analysis. This should be acknowledged in the manuscript, as only the mineral inclusions that constitute natron have been identified (ll. 196-211, and Table 2.) 

→ Done, please see in the modified text

-l. 238 Delete ‘maintained., substitute ‘secured’. 

→ Done, please see in the modified text

-l. 255 ‘posture given to the body’ Suggest ‘position in which the body was arranged’. 

→ Done, please see in the modified text

-l. 265 Delete ‘is’, insert ‘are’ (plural)

→ Done, please see in the modified text 

-l. 352 Delete ‘Univeresety’, insert ‘University’. Misspelling. 

→ Done, please see in the modified text

-l. 359 Ibid. ‘University’

→ Done, please see in the modified text

------------

Reviewer #2

The line numbers are given for the “track change” document.

-Do not capitalise Ancient (as in Ancient Egyptian should be ancient Egyptian).

→ Done, please see in the modified text 

-l. 29 granted completely continuous access is awkward phrasing.

→ Done, please see in the modified text 

-l. 40-46 somewhat unclear in meaning, mainly due to language issues.

→ Done, please see in the modified text 

-l. 50 better if: several individuals representing the same species were to read several individuals of the same species.

→ Done, please see in the modified text 

-l. 53 Instead of mummification preparation protocol the authors should say mummification, or alter the phrasing. Also, awkward phrasing: involving cessation of putrefaction has been brought forward.

→ Done, please see in the modified text 

-l. 61 should say provenanced. Would be useful to have the dimensions of the animal (skull and post-cranial).

→ Done, please see the addition in the modified text 

-l. 64 not driven by…. Choose another word.

→ Done, please see in the modified text 

-l. 66 unclear: although (17)) do mention both Esna…

→ Done, please see in the modified text 

-l. 67-68 no proof that bandaging provides chronological data or provenance data

→ Removed from the text

-l. 68 should read on the period of the…

→ Done, please see in the modified text

-l. 81 maybe better to say dark coloured mummification materials (balm) and then refer to it as balm thereafter

→ Done, please see in the modified text 

-l. 82 Unclear: The limbs are hidden in skin folds

→ Done, please see in the modified text 

-l. 83 Provide more precise details of the location of the incisions. Maybe on an image? Also, break, not brake, and indeed, the whole sentence following from ‘brake’ needs to be revisied for clarity (clearly a problem with language).

→ Done, please see in the modified text and on the revised figure 1

-l. 85-87 Unclear: Fractures into the balm layer and the scattered remains of textile, rope, 85 plants shoots, and other isolated elements indicate that certain parts of the original external layers were removed from the mummy.

→ Done, please see in the modified text and on the revised figure 1

-l. 110 Not sure about anatomical properties

→ Done, please see in the modified text 

-l. 136: Thus, only osteoderms, phalanges, skull preserved, almost all with cut marks? But then l. 137 mentions metapodia. Maybe authors should provide a list of all the bones that appear in fig. 3

→ Done, added in the caption of the figure 3

-l. 146 maybe vegetal or a variety of plant matter rather than floral? It is only a quibble.

→ Done, please see in the modified text 

-l. 157 Abrupt introduction of rope…

→ Done, the rope is mentioned in the introduction part (l.91) of the revised document

-l. 168 Awkward English

→ Done, please see in the modified text 

-l. 181 English issues : dorsal aspects is stick to a remaining part of the cartilaginous

→ Done, please see in the modified text 

-l. 278 When mummies are being produced dermestids can enter the body, as seems to be the case with this animal. Sometimes, even when the animal is covered with natron, these beetles can enter the body and carry out their entire life cycle during the course of desiccation. See: Ikram, S. 2015. ‘Experimental Archaeology: From Meadow to Em-baa-lming Table’, in C. Graves-Brown (ed.) Egyptology in the Present: Experiential and Experimental Methods in Archaeology, pp. 53-74. Swansea: The Classical Press of Wales. The authors have to argue more convincingly that the crocodile had been dead for some time when it was eviscerated and basically turned into a taxidermied crocodile in term of having most of the bones, flesh and organs removed and being stuffed with grassed (but after being traditionally desiccated using natron, and then covered with the conventional balm?).

→ Done, please see in the modified text paragraph 4. Insects 

-l. 290 the skin was cleaned—maybe the authors could suggest with what?

→ Unfortunatly, nothing in the data can bring us solid argument regarding how this cleaning was performed. It could have been purely mechanical or using water, but in both cases, no visible traces could be retrieved. Hence, we prefer not to bring any hypothesis on this specific point.

-l. 293 Possible that the natron was not well cleaned off the body, which is why it stayed on, rather than having the balm mixed with natron. Perhaps analyses of the balm in these areas would clarify the situation. SEM?

→ We propose a mixture of balm with natron because of the 3D repartition of natron inclusions inside the balm. Indeed, the skin was not perfectly cleaned of its natron (we can see a thin layer of natron in direct contact with the skin surface) but it is also possible to see a “direction” where the natron in suspension in the balm “fell” during the drying, due to a decantation process that occurred before the full solidification of the balm. We modified the text to make this explanation clearer.

-l. 300 Palm leaves, particularly the ribs, and textiles are traditionally used to make crocodiles (and sometimes humans) more rigid and stackable. Generally this is more common in immature crocodiles, but is also known in mature animals.

→ We added this point to the text as it is indeed a very common feature in small crocodile mummies.

---

## [Editor Report · Decision Letter 1]

31 Jan 2020

Synchrotron “virtual archaeozoology” reveals how Ancient Egyptians prepared a decaying crocodile cadaver for mummification

PONE-D-19-24542R1

Dear Camille Berruyer,

We are pleased to inform you that your manuscript has been judged scientifically suitable for publication and will be formally accepted for publication once it complies with all outstanding technical requirements.

With kind regards,

William Oki Wong, Ph.D.

Academic Editor

PLOS ONE
---

## [Editor Report · Acceptance letter]

6 Feb 2020

PONE-D-19-24542R1 

Synchrotron “virtual archaeozoology” reveals how Ancient Egyptians prepared a decaying crocodile cadaver for mummification 

Dear Dr. Berruyer:

I am pleased to inform you that your manuscript has been deemed suitable for publication in PLOS ONE. Congratulations! Your manuscript is now with our production department. 

With kind regards,

on behalf of

Dr. William Oki Wong 

Academic Editor

PLOS ONE